# S-Adenosylmethionine (SAMe) for Liver Health: A Systematic Review

**DOI:** 10.3390/nu16213668

**Published:** 2024-10-28

**Authors:** Kyrie Eleyson R. Baden, Halley McClain, Eliya Craig, Nathan Gibson, Juanita A. Draime, Aleda M. H. Chen

**Affiliations:** School of Pharmacy, Cedarville University, Cedarville, OH 45314, USA; halleymcclain@cedarville.edu (H.M.); ecraig@cedarville.edu (E.C.); ngibson@cedarville.edu (N.G.); juanitaadraime@cedarville.edu (J.A.D.); amchen@cedarville.edu (A.M.H.C.)

**Keywords:** S-adenosylmethionine, SAMe, liver, nutraceutical

## Abstract

Background/Objectives: S-adenosylmethionine (SAMe) is a natural compound implicated in the treatment of liver dysfunction. In this systematic review, our objective was to determine the efficacy, safety, and optimal dose of SAMe in liver diseases. Methods: Using the PRISMA methodology, we searched PubMed, CINAHL, and Web of Science using key MeSH search terms. For title/abstract screening, full-text review, and data extraction, two independent researchers reviewed articles, and a third researcher resolved conflicts. Data extraction also included a quality assessment of included articles. Results: Of the 1881 non-duplicated studies, 15 articles focusing on SAMe use in the liver were included. All included studies (*n* = 15) scored a 4 or 5 out of 5 points on the quality assessment, which indicated high study quality. Overall, SAMe was effective in improving liver-related parameters with few adverse events, which were primarily mild, transient gastrointestinal complaints. Conclusions: The most common doses were SAMe 1000 mg or 1200 mg per day with or without another treatment or natural supplement. Future studies are needed to assess long-term efficacy and safety data of SAMe and the optimal route of administration in liver diseases.

## 1. Introduction

The negative impact of liver disease has increased worldwide, ranking as the eleventh leading cause of death globally and the tenth in the United States [1,2]. Of these liver-related deaths, half are attributed to cirrhosis and the other half to both viral hepatitis and hepatocellular carcinoma (HCC) [1]. Death specifically due to liver cancer, such as HCC, has risen at drastic rates, increasing by 43% in U.S. adults 25 years and older between 2000 and 2016 [3]. Furthermore, the prevalence of non-alcoholic fatty liver disease (NAFLD), a leading contributor of chronic liver disease, has also increased alarmingly over the last couple decades, affecting 32% across the world and up to 47.8% in the United States [4,5].

The pharmacological treatment of liver diseases may include antivirals, immunosuppressants, and/or medications for symptom management [6]. However, challenges with these drug therapies can arise due to complications such as increased risk of infection, metabolic abnormalities, and even development of antiviral resistance. Therefore, with the increasing burden of liver disease, there is a need for treatment options that are both safe and effective in managing their complications.

S-adenosylmethionine (SAMe) is a compound that can be delivered in a supplement that is recognized for its positive effects across many physiological systems [7,8]. SAMe is synthesized in the liver from L-methionine and adenosine triphosphate (ATP), playing a crucial role as a primary methyl donor required for numerous biological functions [7]. In addition, it is a known precursor to glutathione, which establishes the antioxidant potential of SAMe in liver injury and disease [9]. The beneficial effects of SAMe in specific liver diseases has not been fully established in humans. However, preliminary studies of SAMe are promising, especially in improving liver parameters in fatty liver disease, hepatitis, and HCC [10]. In the liver, a reduction in SAMe levels affects lipid metabolism, contributing to the development of hepatic steatosis, injury, and even cancer [11]. Furthermore, SAMe has been shown to be reduced in chronic liver diseases, such as hepatic cirrhosis and HCC [9]. However, many available studies of SAMe use for liver diseases in humans are small or have suboptimal methodology, making an updated review of the literature necessary to assess our current understanding of SAMe in the liver. Therefore, we conducted a systematic review to evaluate the safety, efficacy, and optimal dose of SAMe in liver-related diseases.

## 2. Methods

For this systematic review, the PRISMA methodology was used and complied with. Details of the PRISMA checklist can be found in the Appendix A [12]. Aligning with the research objective, an initial search strategy was identified and a research librarian was consulted to refine the search strategy. The following MeSH search terms were identified and utilized: “S-Adenosylmethionine AND Liver”. The research librarian refined and used these terms in PubMed, CINAHL, and Web of Science to determine the breadth and accuracy of the search. The final search terms were reviewed by the research team, and the research librarian performed the search for 1 January 2004–17 April 2024. Zotero (v 7.0, Fairfax, VA, USA) was used to clean the search results (removing 2 retracted articles) before uploading them into a systematic review management software, Covidence (Melbourne, VIC, Australia). Covidence automatically removed any duplicates.

Before beginning the review process, the senior investigator (AC) trained the research team on the software platform (Covidence) and the protocol. Covidence was utilized for the entire systematic review process. At each phase of the systematic review, two individuals independently conducted this step, with the senior investigator serving in the role of resolving conflicts and checking for consistency in protocol application. Two reviewers had to select “yes” for a study to proceed to the next phase or “no” for a study to be excluded. In the first phase, reviewers could select “maybe”, which would allow for a study to proceed forward to full-text review for further review.

Articles were included if they were research studies, written in English, and contained human subjects. Alternatively, articles were excluded if they were not research articles (ex: review articles, expert opinion, commentaries, or guidelines). Key information that articles had to contain related to the study aims were the use of SAMe in the study and the examination of a liver-related condition.

In Phase I, the screening of titles and abstracts was performed in terms of their adherence to the inclusion criteria and study aims. In Phase II, full-text articles were uploaded into Covidence and evaluated. If a study was excluded in this phase, reviewers had to select a reason for exclusion. In Phase III, the included articles underwent data extraction. A template was pre-built in Covidence to collect data. Given the diversity of study designs, quality assessment was performed using the MMAT (Mixed Methods Appraisal Tool) [13]. If data were missing, this was noted in the final tables. Before finalizing the work, the research team reviewed and approved the final data extraction tables.

## 3. Results

From our search, 2207 articles were identified across several databases (PubMed = 521, CINAHL = 173, Web of Science = 1513). Following the elimination of duplicates, 1881 articles were included in this review. Figure 1 shows the PRISMA flow diagram outlining the study methodology, which led to data extraction from 15 articles.

### Study Characteristics

Across the 15 articles reviewed [14,15,16,17,18,19,20,21,22,23,24,25,26,27,28], there were 1799 participants, excluding systematic reviews. In studies that reported gender, 69.7% of participants were male. Regarding the duration of the studies, the median study length was 8 weeks, while the average study length was 21.2 weeks.

Liver-related diseases included in the studies were fatty liver disease (both alcoholic and non-alcoholic), neonatal jaundice, cholestatic liver disease, chronic liver disease, chemotherapy-induced liver disease, hepatitis B virus (HBV)-related HCC, hepatitis C, viral hepatitis, and primary biliary cholangitis. Efficacy was assessed using various measures, including liver function [aspartate aminotransferase (AST), alanine aminotransferase (ALT), lactate dehydrogenase (LDH), total bilirubin (TBil), gamma-glutamyl-transpeptidase (GGT), alkaline phosphatase (ALP), albumin, etc.], liver fat content, liver morphology, and varied inflammatory mediators/indicators.

Table 1 provides an overview and summary of the outcomes. Overall, positive changes in liver markers were found in all 15 studies with minimal to no adverse effects. SAMe dosing in liver diseases ranged from 200 mg to 2400 mg per day, with the most common doses being 1000 mg to 1200 mg per day. Three studies reported titration up to the target dose.

A more comprehensive overview of study characteristics can be found in Table 2. This includes the specific liver disease, interventions, and measurements used to assess liver function. Comparators included placebo, a different nutraceutical (e.g., Si Mo Tang), chemotherapy, or conventional symptomatic treatment.

Of our 15 included studies, there were 3 articles that evaluated alcoholic liver disease, 3 for cholestatic liver disease, 2 for HCC, 2 for viral hepatitis, 2 for chemotherapy-induced liver toxicity, 1 for non-alcoholic liver disease, 1 for neonatal jaundice, and 1 for chronic liver diseases.

For dosing, strategies differed based on the specific liver disease. For alcoholic liver diseases, doses were between 400 mg and 1200 mg per day, with higher doses in severe alcoholic hepatitis. Higher doses of 800 to 2000 mg per day were seen in cholestatic liver disease. HCC was treated with doses of 1000 mg per day with one study increasing to 1500 mg per day after 10 days. In viral hepatitis, initial doses of 800 mg to 1000 mg per day were given, which increased up to 2400 mg per day after 8 weeks in hepatitis C. Chemotherapy-induced liver toxicity was treated with doses of 1200 mg to 1800 mg per day. The smallest doses were used for non-alcoholic liver disease and neonatal jaundice at 200 mg per day and 500 mg per day, respectively. In the article looking at seven different chronic liver diseases, doses between 400 mg and 1200 mg per day (20–30 mg/kg/day) were seen. Five of our included articles evaluated intravenous formulations of SAMe.

The most common measurements of liver function used in the studies were AST and ALT (*n* = 12 studies) and TBil or bilirubin (*n* = 12). Among the other liver measurements, immune function and inflammatory markers were also used.

Table 3 provides the study outcomes results for efficacy, while Table 4 shows safety results. Table 5 shows the quality assessments. All 15 studies included efficacy data and 11 studies included safety data. Improvements in liver function were seen in all 15 studies. SAMe significantly improved liver function in all studies evaluating chemotherapy-induced liver toxicity, HCC, and viral hepatitis. It also demonstrated significant beneficial effects on liver fat content in non-alcoholic fatty liver disease. For alcoholic liver disease, significant improvements were seen in two of the three studies, with one study showing significant improvement only in smooth muscle actin. The combination of SAMe and Si Mo Tang in neonatal jaundice demonstrated an effective rate of 96%, which was significantly higher than either agent alone. For cholestatic liver diseases, significant improvements were seen in both liver function and symptoms of itching.

In the 11 studies assessing safety, gastrointestinal symptoms had the most reported complaints. Five of these studies reported that adverse events were not significantly different between groups and three reported that adverse events were less in SAMe-treated groups.

## 4. Discussion

Based on our results, SAMe appears to exhibit therapeutic advantages in the treatment of liver diseases. Figure 2 illustrates the proposed mechanisms by which SAMe impacts the condition of the liver.

SAMe seemed to be particularly helpful after chemotherapy to improve both liver function and patient outcomes in metastatic colorectal cancer [14,15]. Other liver diseases that have benefited from SAMe use include non-alcoholic fatty liver disease, chronic liver diseases, HCC, and neonatal jaundice [14,15,16,17,18,19]. On the other hand, some studies report that SAMe may not be as beneficial in alcoholic liver disease [20,21]. Nevertheless, its beneficial effects on many hepatic measurements make its application widespread among liver diseases. Furthermore, SAMe may have synergistic effects in combination with other therapies, such as with prednisolone in alcoholic hepatitis or with Si Mo Tang in neonatal jaundice [19,22].

While most studies compared SAMe (or a combination) to placebo, a few directly compared SAMe or its combination to other supplements or conventional treatments. In the study comparing SAMe and Si Mo Tang in neonatal jaundice, there was no statistical significance between the two therapies; however, their combined effects had a significantly higher effective rate than either alone [19]. In viral hepatitis, SAMe also significantly outperformed conventional symptomatic treatment in improving liver function tests and response rates [24]. For combination regimens, the addition of SAMe to prednisolone in severe alcoholic hepatitis significantly increased the rate of responders compared to prednisolone alone [22]. Likewise, adding SAMe to chemotherapy significantly improved a number of assessments, including the need for course delay, grade of liver toxicity, and liver parameters (AST, ALT, TBil, and GGT) [15]. Ursodeoxycholic acid (UDCA), a common treatment for patients with cholestatic liver disease, combined with SAMe 1200 mg per day also showed to be significantly better compared to UDCA alone in achieving a total effective rate [26]. This same combination in primary biliary cholangitis showed significant improvement in fatigue and itch symptoms and, in non-cirrhotic patients, in ALP, GGT and cholesterol levels [27]. Despite the promising results, however, further studies are needed to directly assess SAMe as a monotherapy or in combination with the standard of care in more types of liver diseases.

SAMe was dosed in the range of 200 mg to 2400 mg per day, with the most common doses being 1000 mg or 1200 mg per day. The optimal dose for maximizing efficacy and minimizing adverse effects may depend on several factors, including the type of liver disease and patient-specific considerations, such as gender and age [23]. Based on the current evidence, we recommend starting doses of 800 mg or 1000 mg per day for most liver diseases. For patients with neonatal jaundice, a lower dose of SAMe 500 mg (with Si Mo Tang) appears to be effective. On the other hand, patients with cholestatic liver disease may require higher doses of 1200 mg per day to increase efficacy. If weight-based dosing is desired, a range of 20 to 30 mg per kg per day is a good reference for chronic liver diseases [16]. Considering dosing route, intravenous SAMe may be a helpful option in HCC, cholestatic liver disease, viral hepatitis, and alcoholic liver disease. Nevertheless, the oral route is also commonly used and effective, and it may be less expensive [7]. However, bioavailability of different oral formulations must be considered as it may vary and potentially affect treatment outcomes [7].

Among the studies reviewed, SAMe demonstrated a favorable safety profile. The most frequently reported side effects included gastrointestinal disturbances, such as diarrhea, abdominal discomfort, and nausea. Other studies reported no difference compared to placebo or even less adverse events in the SAMe group [8,17,22,24]. These gastrointestinal effects were generally mild and resolved upon discontinuation of therapy. In one study reporting severe adverse events, the highest dose of SAMe (2400 mg per day) seen in our included studies was used [25]. This supports our recommendation to avoid this higher dose in liver disease [29,30,31,32,33].

Furthermore, one potential concern that has been raised regarding the safety of SAMe is its potential to increase homocysteine levels. After SAMe is demethylated, it is converted to S-adenosylhomocysteine (SAH) and then further hydrolyzed to homocysteine (Hcy) [34]. Homocysteine can either enter the trans-sulfuration pathway to promote glutathione synthesis or convert back to methionine and, subsequently, SAMe [34,35]. The concern with elevated Hcy is primarily related to its link to endothelial cell dysfunction and atherosclerosis [36]. Besides an increased level of SAMe, Hcy can be elevated for several other reasons, such as genetics, B vitamin and folate deficiencies, increased age, and certain drugs and disease states [36]. While the change in Hcy levels was not a primary focus in our review, no adverse effects related to Hcy levels were reported. Two of our studies assessed Hcy levels over 24-week study periods and found that there were no changes in serum concentration in either treatment or placebo groups [21,25]. Importantly, one of these studies required daily supplementation with a multivitamin with B6, B12, and folic acid [25]. Nevertheless, the impact of aforementioned factors on homocysteine warrants further investigation and research.

Our systematic review has several notable strengths. The application of PRISMA methodology enhances the rigor and reliability of our data and design as a systematic review. Moreover, all included studies were rated as high-quality, which further supports the strength of the evidence. Additionally, this review encompasses a broad range of liver diseases, providing an extensive evaluation of SAMe’s therapeutic potential in this area. Nonetheless, this review is limited by the variability in study designs, populations, and dosages. Though it is difficult to generalize findings, data in a wide variety of contexts can be helpful for different applications of SAMe use. Our review process is also not immune to potential human error. Thus, another limitation is any inconsistencies that may have occurred in applying the inclusion and exclusion criteria in our literature search.

Other than in HBV-related HCC, there is a lack of long-term high-quality studies assessing SAMe in liver diseases. In this area, most of the published data have evaluated SAMe for 12 months or less. Therefore, follow-up studies are needed in these patients to understand long-term impact of SAMe even if only a short treatment duration was used. Other current challenges of SAMe use in liver diseases include the inconsistency of improvements in liver measurements in some studies, specifically LFTs and liver morphology in alcoholic liver disease. In addition, the bioavailability of oral SAMe is reported to be 1–2%, which may limit the impact of treatment when given orally. However, newer types of enteric formulations are being developed to help increase systemic absorption [37,38]. Finally, there is a lack of clear clinical guidelines or consensus on the use of SAMe in the treatment of liver diseases, which limits its potential to benefit patients, as well as provide additional data.

Future high-quality studies are needed to assess long-term safety and efficacy of SAMe in liver diseases. Moreover, none of the studies in our review compared SAMe to standard therapy. More large-scale trials across various types of liver disease and compared to standard therapy would be helpful in establishing clear evidence of SAMe’s efficacy and safety. Furthermore, enhancing the bioavailability of oral formulations and comparing them to alternative routes (e.g., intravenous, intramuscular) is necessary to optimize therapy. Understanding the impact of SAMe on routes of administration, specific liver diseases, and various demographics (including age, gender, homocysteine levels, and nutritional status) is crucial to working toward its integration in clinical guidelines.

## 5. Conclusions

The current state of the evidence indicates that SAMe can improve liver function parameters and alleviate disease symptoms. Before using SAMe in this context, health care professionals should consider patient-specific factors, such as the specific liver disease being treated, SAMe formulation and dosage, and goals of therapy. SAMe can serve as a valuable alternative to first-line therapies, especially if they are ineffective, inappropriate, or if the patient wants an effective approach without considerable side effects. Future research is essential to evaluate SAMe’s long-term efficacy and safety and to determine the optimal route of administration based on liver disease.

## Figures and Tables

**Figure 1 nutrients-16-03668-f001:**
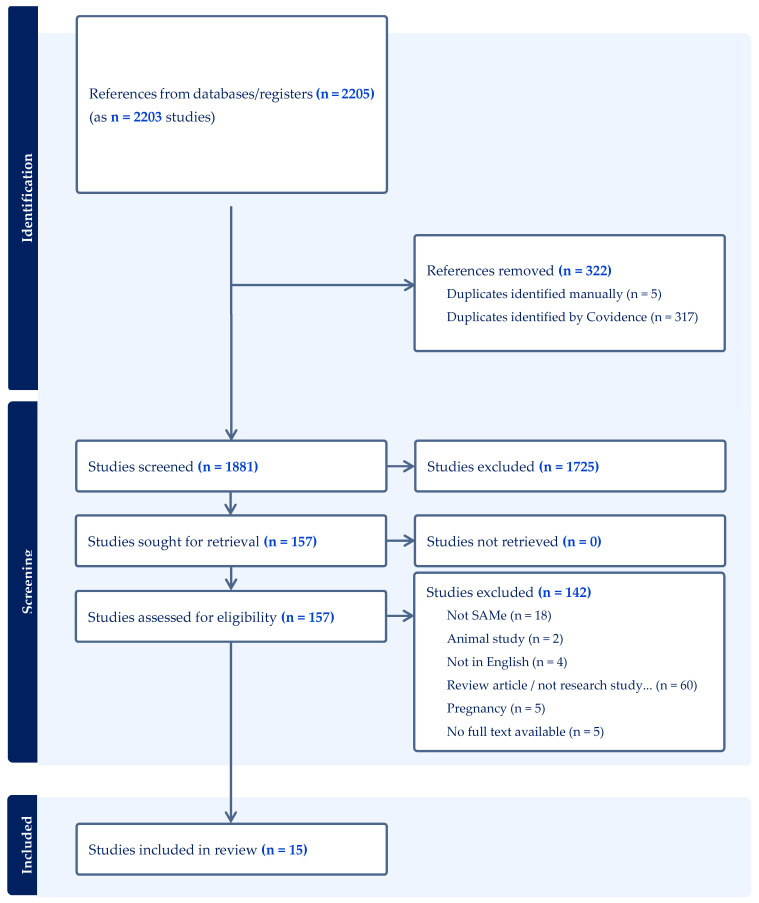
PRISMA [12] diagram overviewing study inclusion and exclusion process.

**Figure 2 nutrients-16-03668-f002:**
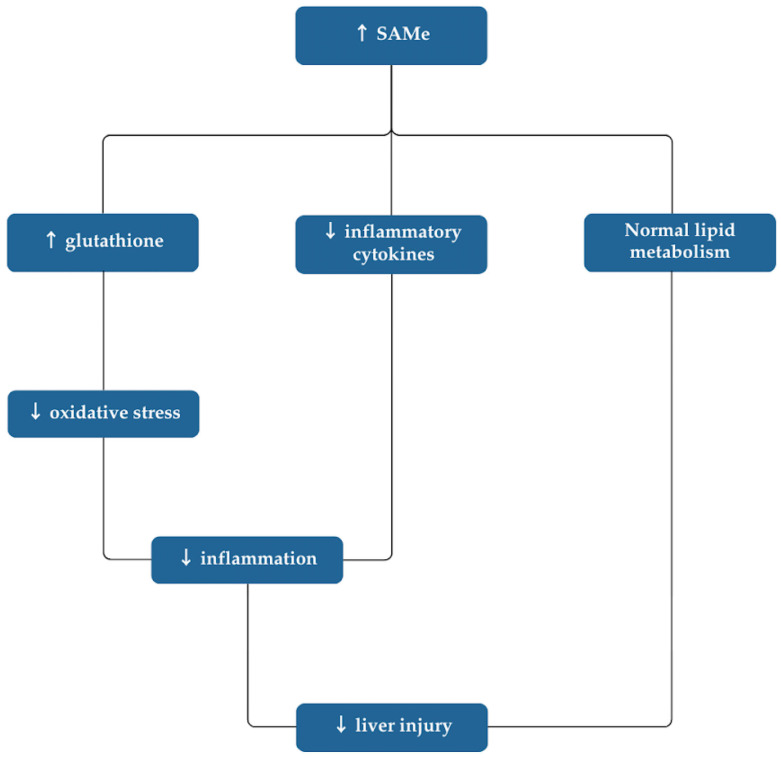
Mechanism of SAMe in liver disease.

**Table 1 nutrients-16-03668-t001:** Overview of study findings related to safety and efficacy.

Condition	Efficacy Summary	Safety Summary	Dosing Ranges
Liver-Related	A total of 15/15 studies found some liver-related benefits in patients taking SAMe.	Serious adverse events were rare and typically not greater than placebo. In some cases, there were lower adverse event rates than placebo. Most common side effects were gastrointestinal.	Range: 200–2400 mg of SAMe per day. Most common doses were 1000 mg or 1200 mg of SAMe per day.

**Table 2 nutrients-16-03668-t002:** Characteristics of included studies examining the role of SAMe in liver diseases.

Author (Year) Study Design | Location N of Patients | Study Length	Intervention (with Dose) and Comparator	Disease	Measurement of Liver Function
Benic (2022) [14] SR *n* = 28 articles (3 articles on SAMe used for analysis) N/A Varying lengths	SAMe: 1200 mg/day in each study	Metastatic colorectal cancer (1 study) Cancer chemotherapy-induced liver toxicity (2 studies)	AST, ALT, LDH, TBil, GGT, and ALP
Ferro (2022) [23] RCT *n* = 140 (127 completed) Europe 12 weeks	Nutraceutical capsule daily (curcumin complex, ω-3 PUFAs, BPF, artichoke leaf extract, black seed oil, pricoliv, GHS, SAMe 200 mg and other natural ingredients) Comparator: placebo	Non-alcoholic fatty liver disease	Liver fat content (CAP score)
Guo (2015) [16] SR/MA *n* = 705 participants across 11 studies N/A Varying lengths	SAMe: 20–30 mg/kg/day (400–1200 mg/day)	Chronic liver diseases	Liver function
Guo (2016) [17] Non-randomized experimental *n* = 697 China 24 months	All: magnesium isoglycyrrhizinate 100 mg/day Group A and C received: SAMe 1000 mg IV (3 days pre-surgery to 7 days post-surgery) then 1500 mg/day Groups B and D received: placebo	HBV-Related HCC Group A, B: Early stage Group C, D: Advanced stage	Liver function
Le (2013) [20] RCT *n* = 14 United States 24 weeks	SAMe: 400 mg/day Comparator: placebo	Alcoholic liver disease	Liver morphology
Li (2022) [19] Non-randomized experimental *n* = 149 China 10 days	Group A: 500 mg SAMe/day Group B: Si Mo Tang Group C: 500 mg SAMe + Si Mo Tang/day	Neonatal jaundice	Liver function, cardiac enzymes, immune function, serum transferrin (TRF), and C-reactive protein (CRP) levels
Liu (2014) [18] RCT *n* = 81 China Up to 5 days	Pre-Treatment: SAMe 1000 mg 2 h before surgery and 5 post-op days Post-Treatment: 1000 mg 6 h after surgery for 5 days Comparator: placebo	HBV-Related HCC requiring resection	ALT, AST, TBil, DBIL
Lu (2020) [26] Non-randomized experimental study *n* = 177 China 1 month	Group A: ursodeoxycholic acid 15 mg/day + SAMe 1200 mg IV per day Group B: ursodeoxycholic acid 15 mg/day + SAMe 800 mg IV per day Group C (Comparator): ursodeoxycholic acid 15 mg/day	Cholestatic liver disease	ALT AST TBil
Medici (2011) [21] RCT *n* = 37 United States 24 weeks	SAMe: 1200 mg/day Comparator: placebo	Alcoholic liver disease	AST, ALT, bilirubin
Morgan (2015) [25] Phase II RCT *n* = 110 United States 24 weeks	SAMe: 800 mg/day (4 weeks) increased to 1600 mg/d (4 weeks) increased to 2400 mg/day (16 weeks) Comparator: placebo	Hepatitis C	Liver: AFP Well-being: SF-36
Qiao (2021) [24] *n* = 137 Non-randomized experimental study China 14 days	SAMe: 1000 mg/day injection Comparator: conventional symptomatic treatment	Viral hepatitis	Albumin ALT AST TBil Liver fiber indicators IL-6 TNF-α
Tkachenko (2016) [22] RCT *n* = 40 Russia 28 days	Prednisolone + SAMe: 800 mg/day IV for 7 days, then 1200 mg/day oral for 8 weeks Outcomes assessed at 28 days Comparator: prednisolone + placebo	Severe alcoholic hepatitis	Response rate Liver enzymes: ALT, AST, GGT, ALP and bilirubin
Vincenzi (2012) [15] Retrospective analysis *n* = 78 Europe Varied length due to length of chemotherapy	Chemotherapy + SAMe: 800 mg/day Comparator: chemotherapy	Metastatic colorectal cancer: oxaliplatin-induced liver toxicity	Course Delays LFTs: AST, ALT, LDH, TBil, GGT
Wunsch (2018) [27] 12 months Non-randomized experimental study *n* = 24 (18 completed) Europe	UDCA + SAMe: 1200 mg/day Comparator: UDCA	Primary biliary cholangitis	ALT AST ALP GGT TBil Albumin INR Total cholesterol
Yang (2021) [28] RCT *n* = 115 China 2 weeks	UDCA + SAMe: 2000 mg/day IV Comparator: UDCA + SAMe 1000 mg/day IV	Cholestatic liver disease	Itching improvement ALT AST TBil TBA IL-12 IL-18

Acronyms: systematic review (SR), meta-analysis (MA), randomized controlled trial (RCT), hepatitis B virus (HBV), hepatocellular carcinoma (HCC), controlled attenuation parameter (CAP), total bile acid (TBA), alanine aminotransferase (ALT), aspartate aminotransferase (AST), alkaline phosphatase (ALP), a-fetoprotein (AFP), lactate dehydrogenase (LDH), total bilirubin (TBil), direct bilirubin (DBIL), gamma-glutamyl-transpeptidase (GGT), international normalized ratio (INR), tumor necrosis factor alpha (TNF-α), interleukin (IL), polyunsaturated fatty acids (PUFAs), bergamot polyphenol fraction (BPF), glutathione (GHS), ursodeoxycholic acid (UDCA).

**Table 3 nutrients-16-03668-t003:** Study efficacy outcomes for studies examining the use of SAMe in liver diseases.

Author (Year)	Efficacy
Benic (2022) [14]	Positive improvement in liver function with SAMe treatment in all three studies (*p* at least ≤ 0.04)
Ferro (2022) [23]	Greater CAP score reduction in the nutraceutical group vs. placebo (−34 ± 5 dB/m vs. −20 ± 5 dB/m, respectively; *p* = 0.045) More improvements seen in the following: Aged 60 years or less Low baseline HDL-C AST reduction Males
Guo (2015) [16]	SAMe significantly decreased AST levels ◦ MD [95% CI] = −16.15 [−24.95, −7.36], *p* = 0.0003) SAMe did not significantly improve ALT levels vs. placebo, except in two comparisons: ◦ MD [95% CI] = 92.27 [48.97, 135.57], *p* < 0.0001 ◦ MD [95% CI] = −32.7 [−53.85, −11.55], *p* = 0.002
Guo (2016) [17]	Positive impact of SAMe on Group C vs. D: ALT and AST levels on postoperative day 1 were significantly lower (323.1 ± 115.2 vs. 397.5 ± 120.4, 173.5 ± 69.8 vs. 229.5 ± 96.7, respectively; *p* < 0.01, both) Negative impact of SAMe on Group C vs. D: ALB levels on postoperative day 1 were significantly higher in subgroup C than subgroup D (33.1 ± 6.7 vs. 27.2 ± 6.1, *p* < 0.01) The main postoperative complications were statistically less in subgroup C than subgroup D (63/235 vs. 79/206, *p* < 0.01).
Le (2013) [20]	SAMe vs. placebo: Only the median smooth muscle actin stain score decreased from 2 to 1 with a *p*-value = 0.027 No significant differences for any other characteristics (*p* > 0.05)
Li (2022) [19]	Group A: effective rate 73.47% Group B: effective rate 78% Group C: effective rate 96% Group C effective rate significantly higher than A and B (*p* < 0.05) No difference between Group A and B (*p* > 0.05)
Liu (2014) [18]	Pre-treatment (SAMe) vs. post-treatment placebo: ALT significantly lower in SAMe pre-treatment than placebo (*p* = 0.003) on Day 5 Pre-treatment SAMe reduced ALT, AST, TBil, DBIL vs. the other two groups (*p* < 0.05) Post-treatment (SAMe) vs. placebo: No significant differences
Lu (2020) [26]	Total effective rate (marked effectiveness + effectiveness) of Group A > Group C (*p* < 0.05) No other “marked” differences between groups Group C had higher ALT, AST, and TBil than A and B Group A (SAMe) had lower ALT, AST, and TBil than B Marked Effectiveness: ALT, AST and TBil levels decreasing by >50% with clinical symptom resolutionEffectiveness: ALT, AST, and TBL decrease by 50–25% with clinical symptom improvement
Medici (2011) [21]	All patients: Reductions in AST scores: *p* < 0.0001 Reductions in ALT scores: *p* = 0.004 Reductions in bilirubin levels: *p* = 0.004 SAMe vs. placebo: no differences between groups
Morgan (2015) [25]	Liver: SAMe: decrease in serum AFP (34.6 to 32.7 ng/mL) Placebo: increase in serum AFP (35.8 to 41.7 ng/mL) Difference between arms = 7.78, *p* = 0.16 Wellbeing: SAMe: improved SF-36 physical, mental score No significant difference between arms (*p* = 0.88, *p* = 0.21) SAMe: significant improvement in SF-36 mental health subdomain when examined over time vs. control (*p* = 0.04)
Qiao (2021) [24]	SAMe vs. comparator: Total response rate of SAMe vs. comparator was 94.74% vs. 77.05% (*p* < 0.05) Liver function tests were improved in SAMe vs. comparator (*p* < 0.05)
Tkachenko (2016) [22]	Liver: Combined treatment with prednisolone and SAMe induced a rapid decrease in bilirubin on the seventh day in comparison to prednisolone only (*p* = 0.022) Response: Rate of responders was significantly higher in the prednisolone plus SAMe group (19 of 20; 95%) than in the prednisolone group (13 of 20; 65%; *p* = 0.044)
Vincenzi (2012) [15]	SAMe: Significantly reduced the need for course delay (19% vs. 43%; *p* = 0.042) Significantly lower grade of liver toxicity (*p* = 0.009) Non-significant reduced need for dose reduction (*p* = 0.051) Non-significant 53.1% response rate vs. 36.9% comparator (*p* = 0.181) AST (*p* = 0.02), ALT (*p* < 0.001), LDH (*p* = 0.008), TBil (*p* = 0.03) and GGT (*p* < 0.001) were significantly lower
Wunsch (2018) [27]	SAMe: Significant improvement in ALP, GGT, and cholesterol levels was seen in non-cirrhotic patients treated with SAMe Fatigue and itch symptom improvement over 24 weeks (*p* = 0.04 and *p* = 0.006, respectively) No other statistically significant changes
Yang (2021) [28]	Significant decrease in itching scores in both groups (*p* < 0.05) Greater decrease in itching with 2000 mg group (*p* < 0.05) LFTs, IL-12, IL-18 significantly lower in observation than control (*p* < 0.05)

Acronyms: mean difference (MD), liver function tests (LFTs), alanine aminotransferase (ALT), aspartate aminotransferase (AST), total bilirubin (TBil), direct bilirubin (DBIL), gammaglutamyl-transpeptidase (GGT), interleukin (IL).

**Table 4 nutrients-16-03668-t004:** Study safety outcomes for studies examining the use of SAMe in liver diseases.

Author (Year)	Safety
Benic (2022) [14]	Not discussed
Ferro (2022) [23]	One person dropped out within 12 weeks due to allergy, one to diarrhea, and three to abdominal discomfort (bloating, pain or cramps) in the nutraceutical group
Guo (2015) [16]	SAMe did not increase the number of adverse events or the death rate compared with the placebo: RR [95% CI] = 0.94 [0.59, 1.52], *p* = 0.81
Guo (2016) [17]	Less postoperative complications in SAMe Group C vs. D (63/235 vs. 79/206, *p* < 0.01)
Le (2013) [20]	Not discussed
Li (2022) [19]	Not discussed
Liu (2014) [18]	No significant differences between groups
Lu (2020) [26]	Less adverse effects in group A No statistically significant differences in diarrhea, malaise, pruritus, and jaundice between groups
Medici (2011) [21]	No severe adverse events reported Four subjects complained of diarrhea in SAMe group, three of abdominal pain/bloating, two of headaches, one of hair loss, dry mouth, and night sweats
Morgan (2015) [25]	Nausea was significantly more common among subjects receiving SAMe Non-significant but higher: constipation, diarrhea, fatigue, and abdominal cramps/pain 8 SAMe discontinued treatment Seven serious adverse events among SAMe
Qiao (2021) [24]	Total incidence of adverse effects: SAMe = 3.95% Comparator = 18.03% *p* < 0.05
Tkachenko (2016) [22]	Hepatorenal syndrome (HRS) occurred in 20% in the prednisolone group (4 of 20 patients) while no HRS cases were registered in the prednisolone plus SAMe group (*p* = 0.035)
Vincenzi (2012) [15]	Not discussed
Wunsch (2018) [27]	Well-tolerated No severe adverse effects
Yang (2021) [28]	No significant differences in safety outcomes (*p* > 0.05)

Regarding study quality assessments, all studies (*n* = 15) achieved a 4 or 5 out of 5 points using the MMAT Score. This indicates a high study quality for all our included articles.

**Table 5 nutrients-16-03668-t005:** Quality assessment of liver studies (*n* = 15 studies).

Author Year	Clear Research Questions	Data Address Question	Total MMAT Score (Out of 5)
Benic (2022) [14]	Yes	Yes	5
Ferro (2022) [23]	Yes	Yes	5
Guo (2015) [16]	Yes	Yes	5
Guo (2016) [17]	Yes	Yes	5
Le (2013) [20]	Yes	Yes	5
Li (2022) [19]	Yes	Yes	5
Liu (2014) [18]	Yes	Yes	5
Lu (2020) [26]	Yes	Yes	5
Medici (2011) [21]	Yes	Yes	4
Morgan (2015) [25]	Yes	Yes	5
Qiao (2021) [24]	Yes	Yes	5
Tkachenko (2016) [22]	Yes	Yes	4
Vincenzi (2012) [15]	Yes	Yes	5
Wunsch (2018) [27]	Yes	Yes	5
Yang (2021) [28]	Yes	Yes	4

## Data Availability

No new data were created outside of what is published in this article. The study protocol, data collection forms, data extracted from included studies, and all other materials used in this review can be made available upon request. No amendments were made to the study protocol.

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
