# Peer review of "S-Adenosylmethionine (SAMe) for Liver Health: A Systematic Review"

_nutrients, 2024, doi:10.3390/nu16213668_

Round 1
Reviewer 1 Report
Comments and Suggestions for Authors
Correct review. As a speaker for SAME more than 20 years ago, I would like to see some comments of the effect of SAME compared to other drugs ort supplements offered to liver diseases pts. This is my single point.
Author Response
Comment 1: Correct review. As a speaker for SAMe more than 20 years ago, I would like to see some comments of the effect of SAMe compared to other drugs or supplements offered to liver diseases pts. This is my single point.
Response 1: We have added a paragraph to the discussion section addressing SAMe compared to other supplements or drugs based on our results.
Reviewer 2 Report
Comments and Suggestions for Authors
In this work, the authors systematically review the available literature to evaluate the safety, efficacy, and optimal dose of SAMe in liver-related diseases. The search was performed in PubMed, CINAHL, and Web of Science scientific databases for recent articles analyzing the topic under study using keywords such as "S-Adenosylmethionine AND Liver" and their combination.
The manuscript deals with a topic of great relevance and potential interest to the scientific community.
It is also well-written and structured. Methodology (search strategy, quality assessment, inclusion criteria, are detailed and described and are appropriate for the Systematic Review. The study's limitations are properly reported, and the conclusion refers to the result.
However, the manuscript requires some modifications; therefore, I am reporting my suggestions below.
1. Figure 1 needs sharpening. Please improve the quality of the figure.
page 3, line 104 – abbreviations AST, ALT, LDH, TBIL, GGT, and ALP have not been explained before
2. Result, pages 4-5 and Tables – please provide references to the cited study.
3. The tables do not seem to comply with the journal's editorial requirements.
4. Table 2, first column, first row -> 3 articles? It is not clear.
- Table 2, footnote: other acronyms like PUFA, BPF, GHS, CAP, UDCA, and others also should be explained. Please check and correct the entire Table 2.
- What is the key by which the sequence of studies in Table 2 is organized? The same is in Table 3-5.
- Table 3 -> TBIL or (like in a footnote) TbiL?
- Discussion – Is it possible to add an explanation of possible mechanisms of action that justify the research results?
- A graphic abstract illustrating the impact of SAMs on the course of liver diseases and the potential mechanism of action would greatly complement the work. It is not necessary, but it would significantly increase the value of work.
Author Response
Comment 1: Figure 1 needs sharpening. Please improve the quality of the figure. page 3, line 104 – abbreviations AST, ALT, LDH, TBIL, GGT, and ALP have not been explained before
Response 1: We have uploaded the figure separately to maintain readability. We included the full name of each liver function parameter before the abbreviation.
Comment 2: Result, pages 4-5 and Tables – please provide references to the cited study.
Response 2: These have been added accordingly.
Comment 3: The tables do not seem to comply with the journal's editorial requirements.
Response 3: It appears that in the transition of downloading the manuscript some of the formatting did not follow. We have attempted to fix this issue using track changes. We are open to any further formatting adjustments by the Nutrients editorial team.
Comment 4: Table 2, first column, first row -> 3 articles? It is not clear.
Table 2, footnote: other acronyms like PUFA, BPF, GHS, CAP, UDCA, and others also should be explained. Please check and correct the entire Table 2.
What is the key by which the sequence of studies in Table 2 is organized? The same is in Table 3-5.
Table 3 -> TBIL or (like in a footnote) TbiL?
Discussion – Is it possible to add an explanation of possible mechanisms of action that justify the research results?
A graphic abstract illustrating the impact of SAMe on the course of liver diseases and the potential mechanism of action would greatly complement the work. It is not necessary, but it would significantly increase the value of work.
Response 4: This was a systematic review with 28 articles but only 3 reporting SAMe. We used the articles on SAMe for the analysis. We updated the table to make this more clear.
We have updated the Table 2 footnote to include all relevant abbreviations.
We have organized these studies alphabetically by first author’s last name.
All abbreviations for total bilirubin have been updated to reflect the footnote version, “TBil.”
We have included a illustration of the impact on SAMe in liver diseases in our discussion section. The file will be uploaded separately to maintain readability.
Reviewer 3 Report
Comments and Suggestions for Authors
Review of the paper entitled “S-Adenosylmethionine (SAMe) for Liver Health: A Systematic Review” by Kyrie Eleyson R. Baden, Halley McClain, Eliya Craig, Nathan Gibson, Juanita A Draime and Aleda M. H. Chen
Based on the review of available scientific evidence, the authors concluded that S-adenosylmethionine (SAMe) can improve liver function parameters and alleviate disease symptoms.
The authors noted that SAMe is a precursor of glutathione. More precisely, SAMe is a precursor of cysteine, one of the three amino acids, along with glutamic acid and cysteine, that make up glutathione. It can be added that cysteine is the amino acid whose concentration limits the rate of glutathione synthesis. On the other hand, SAMe is a methylating agent. This means that the methionine methyl group in SAMe can be transferred to amino or hydroxy groups on a variety of acceptor molecules. After the methyl group is transferred, SAMe is converted to S-adenosylhomocysteine (SAH), which undergoes hydrolysis leading to the formation of homocysteine (Hcy) and adenosine. In the body, Hcy undergoes two types of transformations: remethylation back to SAMe and transsulfuration, the end product of which is the previously mentioned cysteine. If these pathways are impaired for any reason, hyperhomocysteinemia (hHcy) occurs. As is known, elevated homocysteine levels are an independent risk factor for the development of atherosclerosis. The assessment of homocysteine levels in people with vascular diseases, ischemic heart disease, thromboembolic disease or in patients at risk of stroke is an important predictive factor.
The causes of hHcy include a deficiency of folic acid, vitamin B12 and/or vitamin B6. Folic acid and vitamin B12 are necessary for the proper course of Hcy remethylation, and vitamin B6 is necessary for the proper course of Hcy transsulfuration pathway.
Have the authors encountered the problem of adverse effects of SAMe supplementation related to increased Hcy levels during their studies? Are the above-mentioned vitamins also supplemented during SAMe supplementation? Is this problem related to the metabolic fate of SAMe taken into account at all when making decisions about SAMe supplementation.
I would like the authors to briefly address the problem of hHcy during SAMEe supplementation, taking into account the SAMe metabolism.
Author Response
Comment 1: Have the authors encountered the problem of adverse effects of SAMe supplementation related to increased Hcy levels during their studies? Are the above-mentioned vitamins also supplemented during SAMe supplementation? Is this problem related to the metabolic fate of SAMe taken into account at all when making decisions about SAMe supplementation.
I would like the authors to briefly address the problem of hHcy during SAMe supplementation, taking into account the SAMe metabolism.
Response 1: Two of the studies in our present review looked at homocysteine levels and found that these did not change over the 24-week study periods. In addition, only one of these required an additional vitamin supplementation alongside SAMe. In our discussion section, we have included a more detailed description of homocysteine and addressed that there is a need for further research here.